# Being Physically Active Leads to Better Recovery Prognosis for People Diagnosed with COVID-19: A Cross-Sectional Study

**DOI:** 10.3390/ijerph192214908

**Published:** 2022-11-12

**Authors:** Euripedes Barsanulfo Gonçalves Gomide, Lisa Fernanda Mazzonetto, Jéssica Fernanda Corrêa Cordeiro, Daniella Corrêa Cordeiro, Alcivandro de Sousa Oliveira, Evandro Marianetti Fioco, Ana Claudia Rossini Venturini, Pedro Pugliesi Abdalla, Leonardo Santos Lopes Da Silva, Márcio Fernando Tasinafo Júnior, Denise De Andrade, Lucimere Bohn, Dalmo Roberto Lopes Machado, André Pereira Dos Santos

**Affiliations:** 1Claretiano-University Center, São Paulo 14300-900, Brazil; 2College of Nursing, University of Sao Paulo at Ribeirao Preto, Sao Paulo 14040-902, Brazil; 3Anthropometry, Training and Sport Study and Research Group, School of Physical Education and Sport of Ribeirao Preto, University of Sao Paulo at Ribeirao Preto, Sao Paulo 14040-900, Brazil; 4School of Physical Education and Sport of Ribeirao Preto, University of Sao Paulo at Ribeirao Preto, Sao Paulo 14040-900, Brazil; 5Human Exposome and Infectious Diseases Network (HEID), Ribeirao Preto, Sao Paulo 14040-902, Brazil; 6Faculty of Psychology, Education and Sport, University Lusófona of Porto, 4000-098 Porto, Portugal; 7School of Education and Communication, Algarve University, 8005-139 Faro, Portugal

**Keywords:** pandemic, SARS-CoV-2, physical activity, general health, lifestyle, morbidity, mortality

## Abstract

The regular practice of physical activity helps in the prevention and control of several non-communicable diseases. However, evidence on the role of physical activity in mitigating worsening clinical outcomes in people with COVID-19 is still unclear. The aim of this study was to verify whether different levels of physical activity provide protection for clinical outcomes caused by SARS-CoV-2 infection. A cross-sectional study was conducted with 509 adults (43.8 ± 15.71 years; 61.1% female) with a positive diagnosis of COVID-19 residing in Ribeirão Preto, São Paulo, Brazil. Participants were interviewed by telephone to determine the severity of the infection and the physical activity performed. Binary logistic regression was used to indicate the odds ratio (OR) of active people reporting less harmful clinical outcomes from COVID-19. Active people had a lower chance of hospitalization, fewer hospitalization days, less respiratory difficulty and needed less oxygen support. The results suggest that active people, compared to sedentary people, have a lower frequency of hospitalization, length of stay, breathing difficulty and need for oxygen support. These results corroborate the importance of public policies to promote the practice of physical activity, in order to mitigate the severity of the clinical outcomes of COVID-19.

## 1. Introduction

The Severe Acute Respiratory Syndrome of the Novel Coronavirus (SARS-CoV-2) has infected more than 525 million people and caused more than 6 million deaths [1]. Among those infected, approximately 80% of cases are asymptomatic or with mild symptoms [2], others may have a fever, fatigue, dry cough, myalgia, nasal congestion, cold, sore throat, diarrhea [3], shortness of breath and anosmia [4], and the others can present with severe or critical symptoms that can lead to death [2]. In addition, socioeconomic, geographic and structural factors [5,6], and personal characteristics including age [7,8,9] and sex [9,10] can worsen the clinical outcomes of COVID-19. Comorbidities such as diabetes [11,12], obesity [9,10,11,12,13], respiratory diseases [7,10,14,15], cardiovascular [16], cerebrovascular, hepatic [17] and renal dysfunction [17], excessive use of cigarettes [17], cancer diagnosis [18], immunodeficiency [19], immunosenescence [20] and a sedentary lifestyle [10,21] may also contribute to more severe COVID-19 clinical outcomes [5,22,23].

Among the risk factors mentioned above, the scientific evidence linking a sedentary lifestyle with chronic diseases and premature death is very well established. In contrast, being physically active is one of the main determinants of health [23], as it reduces the occurrence and severity of metabolic syndrome, type 2 diabetes mellitus, cardiovascular disease, cancer, stress, depression and anxiety. Additionally, active living improves physical fitness and stimulates the human immune system [24,25], potentiating the pathogenic activity of tissue macrophages and increasing the number of immunoglobulins, neutrophils, natural killer cells, cytotoxic T cells, B cells and anti-inflammatory cytokines [26]. Studies have shown that sedentary people who become physically active benefit from an enhanced immune system which makes them less susceptible to infections. Moderate-intensity physical activity has been shown to be effective against respiratory tract infections [26].

Recognizing the known positive effects of physical activity on immunity, inflammation and infection [24,25], we hypothesize that physically active people may be better protected against negative clinical outcomes induced by SARS-CoV-2 infection [4,7,20]. Thus, the objective of the study was to verify whether physically active people are less likely to have a negative clinical outcome in response to a diagnosis of COVID-19 in a large community sample of Brazilian adults.

## 2. Materials and Methods

### 2.1. Study Design

A cross-sectional observational study was performed; the data collection process took place between 7 June and 27 December 2020. This manuscript followed the guidelines from the Strengthening the Reporting of Observational Studies in Epidemiology (STROBE) [27] and the Checklist for Reporting Results of Internet E-Surveys (CHERRIES) conference list [28].

### 2.2. Sample, Inclusion and Exclusion Criteria

The sample consisted of adults of both sexes, who had a positive diagnosis of COVID-19. Inclusion criteria were: people ≥ 18 years of age, positive diagnosis for COVID-19, living in Ribeirão Preto, state of São Paulo—Brazil. Exclusion criteria were: presence of any conditions of immunological compromise, prolonged use including the use of corticosteroids and/or chemotherapy, transplant patients, presence of neurodegenerative diseases. The sample size calculation considered a prevalence of hospitalizations for COVID-19 of 10.0%, a precision of 3.5% and a confidence interval of 95%, for a finite population of 33,643 infected during data collection, achieving a sample minimum of 500 to 650 participants. The Power and Sample Size Program^®^ version 3.043 (HyLown Consulting LLC, Atlanta, GA, USA) was adopted for sample size calculation.

### 2.3. Data Collection

The Municipal Health Department of Ribeirão Preto provided the names, telephone numbers and e-mails of 33,643 people diagnosed with COVID-19. Randomly, using Microsoft Excel^®^, seven evaluators, trained and supervised by the principal investigator, made 3814 telephone calls. Of this total, after a maximum of three attempts, 647 participants answered the calls. Data from a total of 14 individuals were disregarded, as the participants did not meet the eligibility criteria, and 124 persons did not wish to participate in the study. Thus, 509 people diagnosed with COVID-19 were interviewed and considered eligible (Figure 1).

### 2.4. Procedures and Ethical Aspects

This study was approved by the Research Ethics Committee of the University of São Paulo at Ribeirão Preto College of Nursing (CEP-EERP/USP) (CAAE n. 39645220.6.0000.5393), following the guidelines that regulate research involving human beings according to the Resolution of the National Health Council (CNS) 466/12. Subsequently, the project was sent for consideration by the Municipal Health Department of Ribeirão Preto and approved (official letter 462/2020-SMS-RP Research Project Evaluation Commission). This entity provided personal information (name, telephone contact and e-mail) of people diagnosed with COVID-19. The names were randomized to avoid the risk of bias in the study. Participants were contacted by telephone and invited to participate in the study by answering the questionnaire “Profile of the person diagnosed with COVID-19” and the short version of the International Physical Activity Questionnaire (IPAQ). The free Google Forms^®^ tool was used to prepare the forms and ensure that all mandatory questions were answered. The usability and technical functionality of the forms were tested before completing the questionnaire. All information collected and stored was used without identifying the study participants. Incentives were not offered for participation in the study. Before obtaining any clinical information, all participants verbally consented to participate in the study, which is also in accordance with the Declaration of Helsinki.

### 2.5. Questionnaire “Profile of the Person Diagnosed with COVID-19”

Initially, the researchers designed a pilot version of the questionnaire based on the research goals of the study. Next, the questionnaire was sent to a committee of judges, consisting of three Ph.D. professionals, such as a Kinesiologist, a Nurse, and a Physician, who were familiar with the research objectives of the study. For each question, the specialist responded “I totally disagree”, or “I disagree” or “Indifferent/neutral”, or “I agree” or “I totally agree”. In addition, it was necessary to answer the question “Do you suggest any modification to this question?” Each answer was analyzed by the study researchers, and when two or more experts marked the same alternative, it was accepted by the researchers. The suggestions for some modifications to the question were also taken into account. This approach is consistent with recommended approaches to the establishment of the content validity of questionnaire surveys [29,30,31,32].

The questionnaire had four pages and was grouped into the following parts: Identification (date of the interview, date of birth, gender, and city or state at the time diagnosed with COVID-19), Block 1 (composed of 16 questions about the diagnosis and hospitalization due to COVID-19), Block 2 (contains 17 questions that refer to the period before the diagnosis by COVID-19), Block 3 (has 9 personal questions that refer to the period before the diagnosis by COVID-19), Block 4 (has 8 questions that seek to know how much physical activity was performed in the period before the diagnosis by COVID-19), Block 5 (has 8 questions that seek to know how much physical activity was performed after recovery from COVID-19) [33].

The questionnaire included questions regarding personal information such as weight and height (to calculate body mass index (BMI)), age, sex, profession, family income, education level, marital status, smoking and drinking habits and pre-existing diseases. In addition, addressed the need for hospitalization, how many days a patient remained hospitalized, respiratory difficulties (breathing difficulty is the disorder that can promote the interruption of the respiratory process, causing a feeling of suffocation (dyspnea)) [34], oxygen support, intubation, days intubated and if there was death. Regarding the clinical outcome of death, information was collected through interviews with the family member or closest caregiver. All questionnaire items had the option of “no answer”, “does not apply” and “prefer not to say”. To ensure that all responses were correctly answered and noted, the researcher read what was noted and asked the respondent if it was correct.

### 2.6. International Physical Activity Questionnaire (IPAQ) Short Version

Physical activity was measured using the Brazilian-validated short version of the International Physical Activity Questionnaire (IPAQ-SV) [32,33]. The IPAQ was applied by asking the participant about the level of physical activity in the week prior to the diagnosis of COVID-19. This instrument assesses the domains and intensity of physical activity—including walking and sitting time that an individual performs as part of their everyday lives. The IPAQ groups and conceptualizes the categories, as follows: (a) Sedentary: does not perform any physical activity for a minimum of 10 continuous minutes during the week; (b) Insufficiently active: practices physical activities for a minimum of 10 continuous minutes per week, but not enough to be classified as active. (c) Active—meets the following recommendations: (a) vigorous physical activity: ≥3 days/week and ≥20 min/session; (b) moderate activity or walking: ≥5 days/week and ≥30 min/session; (c) any added activity: ≥5 days/week and ≥150 min/week. (d) Very active—meets the following recommendations: (a) vigorous activity: ≥5 days/week and ≥30 min/session; (b) vigorous activity: ≥3 days/week and ≥20 min/session + moderate activity and/or walking ≥ 5 days/week and ≥30 min/session. For comparison purposes, in this study, participants were grouped into two groups: sedentary (sedentary and insufficiently active) and active (active and very active) [32].

### 2.7. Statistical Analysis

Data were entered in Microsoft Excel^®^ and validated using double key data entry and verification. This procedure was used to ensure the highest accuracy and quality of the data collected. The variables sex, range age, family income, education level, BMI and clinical outcomes of COVID-19 were presented by absolute (n) and relative (%) frequency. Clinical outcomes examined dichotomously were: hospitalization, hospitalization days, breathing difficult, oxygen support, intubation, days of intubation and death. Fisher’s exact test was used to verify the association between sedentary/physically active people and clinical outcomes in people diagnosed with COVID-19. Binary logistic regression was adopted to indicate the odds ratio (OR) of the physically active group reporting mitigated clinical outcomes of COVID-19 when adjusted for confounding variables including, age, family income, education level and BMI. All analyses were performed using SPSS version 20 (IBM Corporation, Armonk, NY, USA), significance level was set to α ≤ 5%.

## 3. Results

A total of 509 individuals participated in the study (women = 61%). Of these, 49.3% were classified as sedentary. The age range was from 18 to 89, with 78% being between 18 and 59 years old and 22% being 60 years old or older. Regarding socioeconomic status and education level, 87% had a family income above BRL 908.00 and 69% had completed high school, respectively. Regarding nutritional status, 72% were overweight or obese. Related to clinical outcomes, 9% required hospitalization, 6.5% were hospitalized for 6 days or more, 7.9% had breathing difficulty, 7.9% required oxygen support, 2% required intubation, 1.8% were intubated for 6 days or more and 1% died (Table 1).

When we performed Fisher’s exact test, no statistically significant association was found among people aged 18 to 59 years, sedentary and active for all clinical outcomes. However, a statistical significant association was found among people aged ≥60 years old, sedentary and active, for the following clinical outcomes: hospitalization, breathing difficulty and oxygen support (Figure 2).

When we performed Fisher’s exact test considering the whole sample (without grouping by age) statistically significant association was found between sedentary and active people for the following clinical outcomes: hospitalization, hospitalization days, breathing difficulty and oxygen support (Figure 3).

After analyzing the binary logistic regression, controlling for confounding variables, it was observed that physically active people had a lower chance of hospitalization (OR 0.440; 95% CI 0.225–0.861; *p* = 0.017), hospitalization days (OR 0.461; CI 95% 0.212–1.000; *p* = 0.050), breathing difficulty (OR 0.444; 95% CI 0.217–0.909; *p* = 0.026) and oxygen support (OR 0.446; 95% CI 0.217–0.914; *p* = 0.027). The same did not occur for the clinical outcomes intubation (OR 0.260; 95% CI 0.053–1.273; *p* = 0.097), days of intubation (OR 3.297; 95% CI 0.656–16.573; *p* = 0.148) and death (OR 0.847; CI 0.847; 95% 0.136–5.257; *p* = 0.858) (Table 2).

## 4. Discussion

### 4.1. Main Findings of the Results

The study sought to verify whether physical activity levels are associated with the clinical outcomes of people diagnosed with COVID-19. The results, without age group classification, indicate that a physically active lifestyle has beneficial effects on the clinical outcomes of hospitalization, hospitalization days, breathing difficulty and oxygen support. However, considering the age group between 18 and 59 years, no difference was found between sedentary and active people. For the age group 60 years and over, an association was found between sedentary and active for the clinical outcomes of hospitalization, breathing difficulty and oxygen support. To the best of our knowledge, this is the first study to examine the relationship between meeting or not meeting physical activity recommendations (active versus sedentary) with different COVID-19 clinical outcomes. The relationships between self-reported physical activity and the different outcomes of COVID-19 studied are as follows:

#### 4.1.1. Frequency of Hospitalization

Our results show that physically active people had a lower risk of hospitalization when compared to sedentary people. Other studies have also reported similar results. Sallis et al. [9] through a retrospective study of 48.440 North Americans showed that the probability of survival from COVID-19 among people who were physically active prior to infection was greater when compared to sedentary peers. Chen et al. [12] found that active individuals had a lower frequency of hospitalization when compared to sedentary people.

The lower frequency of hospitalization observed among the physically active subjects in the present study can be explained by the protective effects conferred by the regular practice of physical activity. For example, reduction of comorbidities and stress [25], and improvement in the functioning of the immune system [26,34], with an increase in the expression of natural killer cells [6,35,36,37], immunoglobulins, anti-inflammatory cytokines, neutrophils, cytotoxic T lymphocyte, immature B lymphocyte and delayed immunosenescence [6,36] have all been reported as outcomes of regular physical activity. Additionally, it is suggested that being physically active can partially neutralize the harmful effect of SARS-CoV-2 binding to the angiotensin-converting enzyme 2 receptor and contribute to the development of a less severe infection, reducing the risk of hospital admission [37,38].

#### 4.1.2. Hospitalization Days

In our study, physically active participants had a shorter hospital stay when compared to sedentary participants. This finding is consistent with the results of a study carried out in South Korea [6]. This association was also shown in other studies [39,40]. So far, there have been no studies on improving the immune system’s response to COVID-19 infection through physical activity. However, there is evidence that regular physical activity prior to infection may decrease the duration and severity of symptoms of viral infections of the respiratory system [41]. Although not conclusive, these results may be related to the levels of cytokines in the body, which play a significant role in immunity and immunopathology [42,43]. Moderate-intensity physical activity performed daily reduces susceptibility to and morbidity from respiratory viral infections by increasing salivary lactoferrin, leukocytes and other immunoprotective agents [38]. People hospitalized for a long time have respiratory and muscle difficulties that can adversely impact recovery time and delay the restoration of their physical condition prior to hospitalization [44].

#### 4.1.3. Breathing Difficulty

Regarding the need for oxygen support, the associations observed in the present study showed that physically active participants need less oxygen support compared to sedentary people. This lower need for oxygen support was also observed in the study by Mistry and Natesan, when they compared sedentary individuals with another who performed moderate-intensity physical activity [44]. In our sample, the sedentary group was more likely to be overweight or obese compared to the active group. A reduced energy expenditure, characteristic of sedentary people and people with a high BMI, has been shown to be associated with serious complications from COVID-19, including sepsis and breathing difficulty [45]. Despite these results, there is still a lack of direct evidence of a causal relationship between physical inactivity and breathing difficulty in people diagnosed with COVID-19.

#### 4.1.4. Oxygen Support

Regarding the need for oxygen support, the associations observed in the present study showed that physically active participants needed oxygen support less frequently than sedentary participants. This was also observed in a prior study that compared moderately active and sedentary groups [44].

Physical activity can help improve immune system response and mitigate viral infection. In addition, being physically active prevents and treats numerous complications associated with COVID-19, such as heart disease, and neurological and metabolic disorders, including the positive effect on the renin–angiotensin system [46]. Reducing the severity of infections, specifically in the lungs, may decrease the severity of clinical outcomes, which may result in patients not requiring oxygen support [9]. However, further studies are needed to better understand the mechanisms linking physical activity and the need for oxygen support in COVID-19 patients.

#### 4.1.5. Intubation

In our study, no difference was found between physically active and sedentary people regarding the need and duration of intubation. A study conducted in South Africa observed that people with moderate to vigorous levels of physical activity had a lower frequency of intubation compared with people with lower levels of physical activity [47]. It should be noted that the number of participants who had to be intubated was reduced and studies with larger samples are needed to confirm this association.

#### 4.1.6. Death

Recent evidence points out that being physically active can reduce the mortality rates of COVID-19 [42]. Although some studies mention that the risk of death is not different between active and sedentary people, the type of exercise can be an important factor in such results [48,49]. Some researchers have observed a lower risk of death from COVID-19 in people who performed aerobic, muscle, and aerobic and muscle training concurrently, when compared with people who performed aerobic and muscle training infrequently [6]. The same occurred in insufficiently active, active and highly active people when compared to inactive people [6] and athletes compared to non-athletes [7]. In our study, no statistically significant association was found between physically active and sedentary people regarding the frequency of deaths. It is important to highlight that during this research, only the B1.617.2 (delta) variant was found in Ribeirão Preto city. There were no different COVID-19 variants, including AY.4 (delta), AY.39 (delta), B.1.1.7 (alpha), P.1 (gamma), P.1.4 (gamma), P.1.17 (gamma) and P1.1.12 (gamma), [49] during data collection.

The results found may change over time due to the continuity of the pandemic and the emergence of new variants.

### 4.2. Study Limitations

Despite the promising results obtained in this study, some limitations are present and must be considered. The observational and cross-sectional design of the study does not clearly allow for a cause-and-effect relationship [49] between sedentary and active people, and clinical outcomes related to COVID-19. Additionally, our study was unable to explain the association between clinical outcomes of intubation, days of intubation and death, due to the insufficient number of each subsample. In our study, ten patients (2%) of the entire sample were intubated and five (1%) died. Since this study was carried out among participants from a single municipality, located in the southeastern region of Brazil, the results cannot be immediately generalized to other locations with different socioeconomic, demographic and structural factors [5]. As observed in Fisher’s exact test, the age group ≥60 years old is an important factor that influences the clinical outcomes of COVID-19. However, considering the whole sample and controlling important variables into logistic regression (including age group), physical activity stayed significant. Being active is a behavior that can be daily modifiable and should be incorporated in all ages. The IPAQ, which is a measurement of physical activity, is a self-reported questionnaire, and recall bias may occur. Therefore, the results must be interpreted with caution. Future studies, with more age subgroups and less difference among subjects, are needed to eliminate the risk of bias, confirm causality, and explore the relationship between sedentary and active behavior and clinical outcomes in people diagnosed with COVID-19.

### 4.3. Strength of Evidence

Our investigation is one of the few to examine physical activity and clinical outcomes of COVID-19 in Brazil using telephone interviews. The present study was conducted prior to the initiation of the COVID-19 vaccination in Brazil, diminishing the risk of bias related to vaccination status. The International Physical Activity Questionnaire used in this study is easy to apply, has been widely used, has been validated in different populations and countries, including Brazil [50,51], and correlates with direct measures of physical activity [52]. Additionally, the questionnaire that verified the personal and clinical characteristics of COVID-19 patients was validated relative to the objectives of this study [52,53].

## 5. Conclusions

Physically active people have a better clinical prognosis regarding the outcomes of COVID-19, especially with lower frequency of hospitalization, hospitalization days, respiratory difficulty and oxygen support. Our study provides evidence that active behavior is a modifiable risk factor for COVID-19 clinical outcomes. The global effort to encourage the population to adopt a physically active lifestyle, and avoid a sedentary lifestyle, may be a promising strategy for protecting and mitigating the severity of the clinical outcomes of COVID-19.

## Figures and Tables

**Figure 1 ijerph-19-14908-f001:**
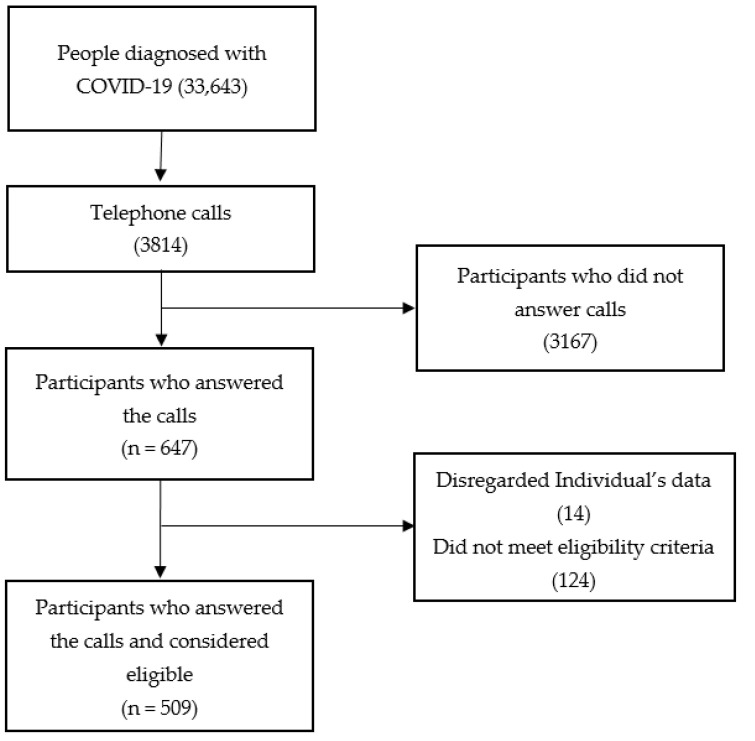
Flowchart of study design. Patients with coronavirus disease (COVID-19). The figure represents a flow diagram of the study.

**Figure 2 ijerph-19-14908-f002:**
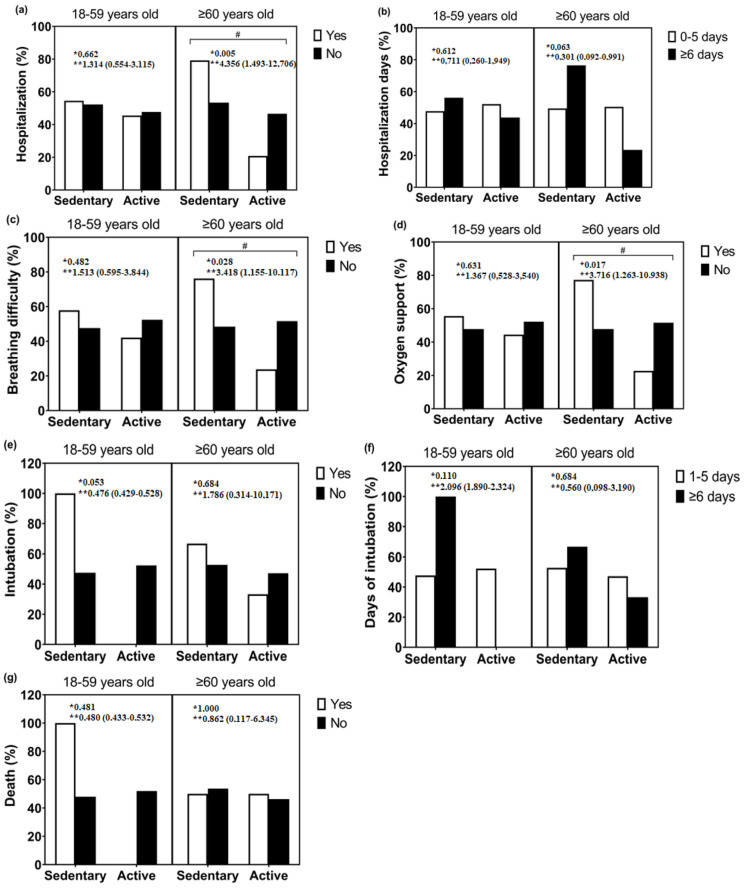
Association between sedentary and active groups, age groups of 18 to 59 years and 60 years and over, and clinical outcomes of COVID-19; (**a**) hospitalization; (**b**) hospitalization days; (**c**) breathing difficulty; (**d**) oxygen support; (**e**) intubation; (**f**) intubation days; (**g**) death. * *p* value; ** OR (IC 95%); #: significant difference between sedentary and active.

**Figure 3 ijerph-19-14908-f003:**
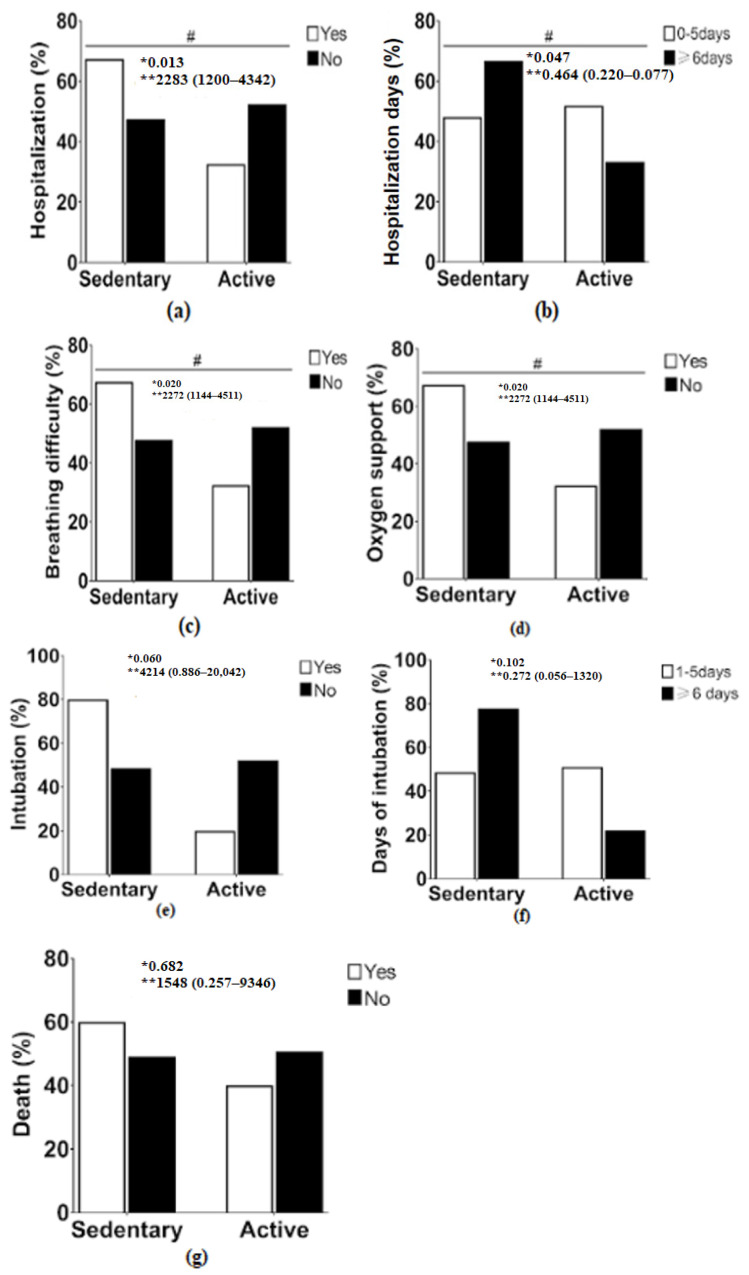
Association between sedentary and active groups and clinical outcomes of COVID-19; (**a**) hospitalization; (**b**) hospitalization days; (**c**) breathing difficulty; (**d**) oxygen support; (**e**) intubation; (**f**) intubation days; (**g**) death. * *p* value; ** OR (IC 95%); #: significant difference between sedentary and active.

**Table 1 ijerph-19-14908-t001:** Personal characteristics and clinical outcomes of the total sample, and grouped into sedentary and active. Ribeirão Preto, Brazil, 2022.

Variables and Personal Characteristics		Sedentary = 251 *n* (%)	Active = 258 *n* (%)	Total = 509 *n* (%)
Sex	Male	102 (40.6)	96 (37.2)	198 (38.9)
	Female	149 (59.4)	162 (62.8)	311 (61.1)
Age grouping	18 to 59 years old	191 (76.1)	206 (79.8)	397 (78.0)
	60 years or older	60 (23.9)	52 (20.2)	112 (22.0)
Family income	Up to 908.00	20 (8.0)	45 (17.4)	65 (12.8)
	Above 908.00	231 (92.0)	213 (82.6)	444 (87,2)
Level of education	Up to full medium	162 (64.5)	189 (73.3)	351 (69.0)
	Higher and postgraduate	89 (35.5)	69 (26.7)	158 (31.0)
BMI (kg/m^2^)	Normoponderal (up to 24.9 kg/m^2^)	62 (24.7)	82 (31.8)	144 (28.3)
	Overweight or obesity (25 kg/m^2^ or more)	189 (75.3)	176 (68.2)	365 (71.7)
Clinical outcomes				
Hospitalization	Yes	31 (12.4)	15 (5.8)	46 (9.0)
	No	220 (87.6)	243 (94.2)	463 (91.0)
Hospitalization days	0 to 5 days	229 (91.2)	247 (95.7)	476 (93.5)
	6 days or more	22 (8.8)	11 (4.3)	33 (6.5)
Breathing difficulty	Yes	27 (10.8)	13 (5.0)	40 (7.9)
	No	224 (89.2)	245 (95.0)	469 (92.1)
Oxygen support	Yes	27 (10.8)	13 (5.0)	40 (7.9)
	No	224 (89.2)	245 (95.0)	469 (92.1)
Intubation	Yes	8 (3.2)	2 (0.8)	10 (2.0)
	No	243 (96.8)	256 (99.2)	499 (98.0)
Intubation days	0 to 5 days	244 (97.2)	256 (99.2)	500 (98.2)
	6 days or more	7 (2.8)	2 (0.8)	9 (1.8)
Death	Yes	3 (1.2)	2 (0.8)	5 (1.0)
	No	248 (98.8)	256 (99.2)	504 (99.0)

**Table 2 ijerph-19-14908-t002:** Odds ratio between sedentary and active groups for the occurrence of clinical outcomes of COVID-19. Ribeirão Preto, Brazil, 2022.

	Variables	Wald	Odds Ratio	Confidence Interval (95%)		*p* Value
Hospitalization	Sedentary/Active	5.649	0.440	0.225	0.861	0.017
	Age grouping	18.271	4.148	2.160	7.964	<0.001
	Family income	0.205	0.810	0.325	2.017	0.651
	Level of education	0.467	0.763	0.351	1.658	0.494
	BMI (kg.m^2^)	1.130	1.556	0.689	3.516	0.288
Hospitalization days	Sedentary/Active	3.837	0.461	0.212	1.000	0.050
	Age grouping	12.256	0.262	0.124	0.554	<0.001
	Family income	0.925	1.626	0.604	4.381	0.336
	Level of education	0.018	1.062	0.442	2.550	0.894
	BMI (kg.m^2^)	0.662	0.679	0.268	1.724	0.416
Breathing difficulty	Sedentary/Active	4.934	0.444	0.217	0.909	0.026
	Age grouping	14.969	3.884	1.953	7.723	<0.001
	Family income	0.128	0.838	0.318	2.209	0.721
	Level of education	0.823	0.676	0.290	1.575	0.364
	BMI (kg.m^2^)	1.816	1.876	0.751	4.681	0.178
Oxygen support	Sedentary/Active	4.863	0.446	0.217	0.914	0.027
	Age grouping	17.673	4.394	2.204	8.762	< 0.001
	Family income	0.114	0.846	0.320	2.239	0.736
	Level of education	0.696	0.697	0.298	1.628	0.404
	BMI (kg.m^2^)	1.715	1.846	0.738	4.619	0.190
Intubation	Sedentary/Active	2.762	0.260	0.053	1.273	0.097
	Age grouping	4.973	4.568	1.202	17.356	0.026
	Family income	0.035	1.226	0.143	10.485	0.853
	Level of education	0.189	0.696	0.136	3.559	0.664
	BMI (kg.m^2^)	0.807	2.620	0.321	21.412	0.369
Intubation days	Sedentary/Active	2.098	3.297	0.656	16.573	0.148
	Age grouping	6.289	0.154	0.036	0.665	0.012
	Family income	0.011	0.891	0.102	7.784	0.917
	Level of education	0.015	1.112	0.210	5.881	0.901
	BMI (kg.m^2^)	0.546	0.450	0.054	3.743	0.460
Death	Sedentary/Active	2.098	3.297	0.656	16.573	0.148
	Age grouping	6.289	0.154	0.036	0.665	0.012
	Family income	0.011	0.891	0.102	7.784	0.917
	Level of education	0.015	1.112	0.210	5.881	0.901
	BMI (kg.m^2^)	0.546	0.450	0.054	3.743	0.460

## Data Availability

The dataset supporting the conclusions of this article is included in the article. Original data are available by request from the authors.

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
