# Peer review of "Being Physically Active Leads to Better Recovery Prognosis for People Diagnosed with COVID-19: A Cross-Sectional Study"

_ijerph, 2022, doi:10.3390/ijerph192214908_

Round 1
Reviewer 1 Report
Dear authors, in the attached document you can find the considerations and suggestions for improving your work.
Best regards.

Author Response
Reply to reviewer attached.

Reviewer 2 Report
Dear Authors,
Congratulations on your work. I have one minor comment. Please provide a flow diagram for your manuscript.
Author Response
Reply to reviewer attached.

Reviewer 3 Report
The cross-sectional study reported active participants have a good prognosis for patients with COVID-19. Some points should be improved.
line 137; please provides the definition of respiratory difficulties.
Regarding the table 1; please provides the cutpoint of overweight or obesity, why did the authors make a cut point for hospitaization days at 0-5 days and 6 days or more, also the intubation days (i.e., 0-5 days vs. 6 days or more)
Table2; physical active people were not a significant difference in days of hospitalization (P=.005). please check!
Study limitaion: line 301-314
One limitation is that the IPAQ which is a measurement of physical activity is a self reported questionnaires (even if the authors clamed that it is easy to use). It might be a self-bias such as recall bias.
Author Response
Reply to reviewer attached.
